# Assessing the consistency of satellite derived upper tropospheric humidity measurements

Lei Shi [1], Carl J. Schreck III [2], Viju O. John [3], Eui-Seok Chung [4], Theresa Lang [5,6], Stefan A. Buehler [5], Brian J. Soden [7]

[1]NOAA NESDIS National Centers for Environmental Information (NCEI), Asheville, NC, USA
[2]Cooperative Institute for Satellite Earth System Studies (CISESS), North Carolina State University, Asheville, NC, USA
[3]EUMETSAT and Met Office Hadley Centre, Exeter, UK
[4]Korea Polar Research Institute, Incheon, South Korea
[5]Meteorological Institute, Universität Hamburg, Germany
[6]International Max Planck Research School on Earth System Modelling, Max Planck Institute for Meteorology, Hamburg, Germany
[7]Rosenstiel School of Marine and Atmospheric Science, University of Miami, Miami, FL, USA

*Correspondence to*: Lei Shi (lei.shi@noaa.gov)

**Abstract.** Four upper tropospheric humidity (UTH) datasets derived from satellite sounders are evaluated to assess their consistency as part of the activities for the Global Energy and Water Exchanges (GEWEX) water vapor assessment project. The datasets include UTH computed from brightness temperature measurements of the $183.31 \pm 1$ GHz channel of the Special Sensor Microwave – Humidity (SSM/T-2), Advanced Microwave Sounding Unit-B (AMSU-B), and Microwave Humidity Sounder (MHS), and from channel 12 of the High-Resolution Infrared Radiation Sounder (HIRS). The four datasets are generally consistent in the interannual temporal and spatial variability of the tropics. Large positive anomalies peaked over the central equatorial Pacific region during El Niño events in the same phase with the increase of sea surface temperature. Conversely, large negative anomalies were obtained during El Niño events when the tropical domain average is taken. The weakened ascending branch of the Pacific Walker circulation in the western Pacific and the enhanced descending branches of the local Hadley circulation along the Pacific subtropics largely contributed to widespread drying areas and thus negative anomalies in the upper troposphere during El Niño events as shown in all four datasets. During a major El Niño event, UTH had higher correlations with the coincident precipitation (0.60 to 0.75) and with 200 hPa velocity potential (-0.42 to -0.64) than with sea surface temperature (SST) (0.37 to 0.49). Due to differences in retrieval definitions and gridding procedures, there can be a difference of 3-5% UTH between datasets on average, and larger magnitudes of anomaly values are usually observed in spatial maps of microwave UTH data. Nevertheless, the tropical-domain averaged anomalies of the datasets are close to each other with their differences being mostly less than 0.5%, and more importantly the phases of the time series are generally consistent for variability studies.

## 1 Introduction

The Global Energy and Water Exchanges (GEWEX) project's water vapor assessment (G-VAP) is organized by the GEWEX Data and Assessments Panel. Three Global Climate Observing System (GCOS) Essential Climate Variables on water vapor are assessed in the G-VAP project, including total column water vapor, upper tropospheric humidity (UTH), and water vapor and associated temperature profiles. The present study is part of the G-VAP activities, focusing on the consistency assessment among satellite-derived UTH measurements.

Measurement of UTH has traditionally been obtained from global radiosonde observations as part of the atmospheric water vapor profiles (e.g., Durre et al. (2018), Ferreira et al. (2019), Brogniez et al. (2015)). In the satellite era, operational routine satellite infrared measurements of UTH started with the High-Resolution Infrared Radiation Sounder (HIRS) instrument onboard Television InfraRed Observation Satellite N (TIROS-N), which was launched in 1978, and the measurement has been continuously produced from the National Oceanic and Atmospheric Administration (NOAA) and Meteorological Operational satellite (Metop) polar orbiting satellites to the present. UTH measurements from geostationary observations have been generated since 1983. Then microwave sounding measurements have been added to the suite of UTH observations since 1991. UTH can also be derived from the new generation hyper-spectral sounders including Atmospheric Infrared Sounder (AIRS), Infrared Atmospheric Sounding Interferometer (IASI), and Cross-track Infrared Sounder (CrIS), and other satellite instruments such as Sondeur Atmosphérique du Profil d'Humidité Intertropicale par Radiométrie (SAPHIR). These satellite sounder measurements complement each other in providing a long-term full picture of the UTH field.

The development of UTH datasets and the examination of temporal and spatial variabilities of UTH have been presented in numerous studies, including both infrared datasets (Soden and Bretherton, 1993; Jackson and Bates, 2001; Brogniez et al., 2006; Shi and Bates, 2011; Iacono et al., 2003; Chung et al., 2007; Gierens et al., 2014; Schröder et al., 2014; Gierens et al., 2018) and microwave datasets (Brogniez and Pierrehumbert, 2006; Chung et al., 2013; Sohn et al., 2000; Buehler et al., 2008; Lang et al., 2020; Brogniez et al., 2015; Moradi et al., 2016). The variability of UTH is regulated by the large-scale atmospheric circulation. The spatial patterns of UTH measurement are highly correlated with widely used climate indices such as the Niño 3.4, Pacific Decadal Oscillation (PDO), Pacific–North American (PNA), and North Atlantic Oscillation (NAO) indices (Shi and Bates, 2011; Shi et al., 2018). The measurements have been applied in various atmospheric variability studies. For example, UTH datasets facilitated studies that showed strong relationship between UTH and El Niño-Southern Oscillation (ENSO) (Mccarthy and Toumi, 2004; Bates et al., 1996; Soden and Lanzante, 1996). UTH was closely associated with deep convection and the evolution of large-scale weather systems (Soden and Fu, 1995; Brogniez et al., 2009; Zelinka and Hartmann, 2009; Luo et al., 2007; Schreck et al., 2013) and interacting with tropical cirrus life cycle (Luo and Rossow, 2004). The measurements have been used in the studies on the strengthening of the Hadley and Walker circulations (Sohn and Park, 2010), the widening of the tropical width (Mantsis et al., 2017), and a possible expansion of the sub-tropical dry zones (Tivig et al., 2020). The UTH datasets facilitated the evaluation of climate models and contributed to a better understanding of large-scale atmospheric processes (Allan et al., 2003; Soden et al., 2005; Chung et al., 2016; Allan

et al., 2022; John et al., 2021). The UTH measurements from both microwave and infrared sounders are used together with ground-based observations and climate model simulations to examine global-scale changes in water vapor and response to surface temperature variability (Allan et al., 2022).

Water vapor is an important greenhouse gas. Its concentration in the free troposphere is controlled by condensation at the cold point and subsequent advection. This leads to a roughly constant relative humidity, which implies a strong increase in absolute humidity content with warming (Soden et al., 2005; Chung et al., 2014). This well understood overall picture is modulated by subtle changes in the distribution of humidity, as measured by the UTH, linked to changes in atmospheric dynamics with warming (Held and Soden, 2000).

Inter-comparison of independently generated UTH datasets provides verification of the datasets' credibility for their uses in research and long-term monitoring. An earlier consistency study (Chung et al., 2016) analyzed UTH derived from HIRS, Advanced Microwave Sounding Unit-B (AMSU-B) / Microwave Humidity Sounder (MHS), and AIRS, and showed that all three products exhibit consistent spatial and temporal patterns of interannual variability. The first phase of the GEWEX UTH assessment (Schröder et al., 2017) included UTH derived from both polar orbiting HIRS, AMSU-B/MHS, and geostationary MVIRI/SEVIRI. Since then, two new polar-orbiting satellite microwave UTH datasets have been developed, and there are now new versions and extended records available for the HIRS and the microwave dataset examined previously. In this study we provide an update on the inter-comparison of the polar-orbiting satellite UTH datasets by including four participating datasets, two of which are new datasets and two of which have updated versions and extended time series.

## 2 Datasets

The four datasets analyzed in this study include UTH generated by the Satellite Application Facility on Climate Monitoring (CMSAF), the Fidelity and Uncertainty in Climate data records from Earth Observations (FIDUCEO) project, the National Centers for Environmental Information (NCEI), and University of Miami (UMIAMI). Three of these are based on microwave sounder measurements, and one is based on infrared sounder measurements. The CMSAF and UMIAMI datasets are derived from AMSU-B/MHS measurements. The FIDUCEO dataset adds Special Sensor Microwave Humidity (SSM/T-2) to the microwave measurements that extends the time series back to 1994. The NCEI UTH data are derived from HIRS Channel 12 measurements. The following provides details of the four datasets.

### 2.1 CMSAF UTH

The microwave sounder UTH data (version 1.0) are derived from AMSU-B and MHS from the $183.31 \pm 1$ GHz channel (John, 2019). The dataset is based on a microwave humidity sounder dataset record generated by EUMETSAT within the framework of the ERA-Clim2 project. A combination of methods was used to estimate inter-satellite biases for the microwave humidity sounders (John et al., 2013; Saunders et al., 2013). There is a simple linear relationship between

brightness temperature (Tb) emanated from water vapor emissions in the upper troposphere and the natural logarithm of
       UTH (e.g., see Buehler and John (2005)), which is the Jacobian weighted relative humidity in the upper troposphere:

ln(UTH) = a + b * Tb                                                                                                     (1)

       The coefficients a and b are determined by linear regression, using a training data set of atmospheric temperature and
       humidity profiles, in which a = 23.467520 and b = -0.099240916.

The CM-SAF UTH is derived for individual pixels and then gridded. The product is provided to users on a global, daily 1.0°
       x 1.0° latitude-longitude grid. UTH is retrieved for all cloud and surface cleared and limb-corrected brightness temperatures
       for each day. These are then separated for ascending and descending passes and binned into each 1.0° grid cell. The time
       series analyzed in this report covers 1999 - 2019 for the CMSAF data.

## 2.2 FIDUCEO UTH

The FIDUCEO UTH (version 1.2) is based on the FIDUCEO fundamental climate data record of recalibrated microwave
       sounder brightness temperatures (Hans et al., 2019), covering the sensors SSM/T-2, AMSU-B and MHS. It uses a new UTH
       definition (Lang et al., 2020) based on the concept that the atmospheric emission layer for a water vapor channel is bounded
       by two characteristic amounts of water vapor integrated from the top of the atmosphere downwards. Using this idea, UTH is
       defined as the mean relative humidity in a layer between two altitude levels z(IWV1) and z(IWV2), at which the integrated
water vapor (IWV) exceeds two viewing angle dependent thresholds IWV1 and IWV2. The thresholds IWV1 and IWV2
       play a similar role in capturing the atmospheric emission layer as the Jacobian in the traditional definition. The IWV
       thresholds were optimized in such a way that the linear relationship between the Tb and the logarithm of UTH is best
       fulfilled for the European Centre for Medium-Range Weather Forecasts (ECMWF) training atmospheres. The data record
       covers the time between 1994 and 2017, and provides monthly mean brightness temperatures and derived UTH along with
estimates of measurement uncertainty on a $1° × 1°$ latitude-longitude grid covering the tropical region (30°S to 30°N). The
       UTH is first derived for individual pixels before gridding. Only pixels close to the nadir view of the satellite are selected.

## 2.3 NCEI UTH

       The NCEI UTH dataset is based on version 3.2 of HIRS channel 12 brightness temperature data (Shi and Bates, 2011).
       Because an infrared sounder cannot sense through clouds, cloudy pixels are removed from the dataset. The cloud-filtered and
limb-corrected channel brightness temperatures are calibrated using derived adjustment coefficients from matched
       overlapping HIRS data between satellites. In this study the UTH is calculated based on the relationship between UTH and
       HIRS channel 12 brightness temperatures centered at 6.7 µm ($T_{6.7}$) derived by Soden and Bretherton (1996):

$$UTH = \frac{cos\theta}{p_0} e^{(a+ b * T_{6.7})}$$                                                                       (2)

       in which $\theta$ is the viewing angle. The $p_0$ is the pressure of the 240 K isotherm divided by 300 hPa ($p_0 = p_{[T = 240 K]}$ / 300 hPa)
which is determined from a training set of ECMWF profiles for 1986-1989 as a function of month, latitude, and longitude.

The coefficients a and b are determined based on the training profiles and radiative transfer model simulation of $T_{6.7}$, in which a = 31.5 and b = -0.115. HIRS UTH dataset has a monthly coverage based on clear-sky observations with a spatial resolution of 2.5° x 2.5° degrees. The UTH is computed from gridded brightness temperature data. The data analyzed in this report cover the period of November 1978 – December 2020.

**2.4 UMIAMI UTH**

The UMIAMI data (Chung et al., 2013) are available as gridded brightness temperatures from AMSU-B and MHS on a 1.5° × 1.5° latitude-longitude grid. Biases due to the difference in local observation time between satellites and spurious trends associated with satellite orbital drift are diagnosed and adjusted for using synthetic radiative simulations based on the Interim European Centre for Medium-Range Weather Forecasts Re-Analysis (ERA-Interim) and ERA5. The adjusted, raincloud-

filtered, and limb-corrected brightness temperatures are then intercalibrated using zonal-mean brightness temperature differences. In this study the formula as that is used by the CMSAF dataset is applied to compute UTH. However, unlike the computation of the CMSAF UTH in which the UTH is first derived for each individual pixel before gridding, the UMIAMI UTH is computed from gridded averaged brightness temperature values. The time series for this study covers 1999 – 2020.

**3 Results and discussions**

The assessment examines several aspects of the UTH datasets, including consistency in time series, spatial feature consistency, and changes during the datasets' common period. The following describes the analyses and results.

**3.1 Intercomparison of time series**

The UTH datasets are most often used to monitor tropical atmospheric activities (e.g., Brogniez et al. (2015), Tivig et al. (2020) and John et al. (2021)). Therefore, the assessment focuses on the consistency of the tropical data. Figure 1 plots the

time series of UTH datasets averaged over the domain 20°S–20°N. These include UTH derived from both microwave 183.31 ± 1 GHz brightness temperatures and infrared 6.7 µm brightness temperatures. Figure 1a displays domain-averaged monthly mean values of UTH, Figure 1b shows the corresponding anomalies, and Figures 1c and 1d show the differences in UTH and in anomaly values, respectively, relative to the values of UMIAMI. In the anomaly calculation, the period 2000-2015 is used for climatology. Figure 1e displays the time series of the Oceanic Niño Index (ONI) (available at

https://origin.cpc.ncep.noaa.gov/products/analysis_monitoring/ensostuff/ONI_v5.php; accessed June 16, 2022). The ONI is constructed using the three-month running average sea surface temperature (SST) anomalies in the Niño 3.4 region (5°S–5°N, 120°W-170°W) (originally presented by Trenberth (1997)).

In Figure 1a, the four datasets appear to be separated with two groups of similar UTH values. The values of CMSAF and FIDUCEO UTH are larger than the values of NCEI and UMIAMI UTH. Among the datasets, the UTH of CMSAF and

FIDUCEO is first computed for each pixel before taking grid averages. For the UMIAMI and HIRS dataset, the gridding of

the brightness temperature is done first, then UTH is computed from averaged brightness temperatures. Based on a study by John et al. (2006), such different ways of computing UTH can lead to a difference of up to 6% UTH due to the non-linearities in the formula that calculates UTH from brightness temperature values. Figure 1c shows that there is a difference of approximately 3-5% UTH between two groups of UTH datasets when a tropical average is taken. In spite of this structural discrepancy, the anomaly plot of the UTH in Figure 1b shows consistency in seasonal and interannual variability patterns among the four datasets. All four datasets show major peaks and dips along the time series in the same phases, though there are differences in the magnitudes of the fluctuations. In the FIDUCEO dataset, SSM/T-2 data before 1998 were at times sparse or missing, causing a few data gaps and some uncertainty in monthly means. Despite different definitions and ways of computing UTH, the anomalies of the four datasets are close to each other.

To quantify the differences between datasets, the relative differences are calculated. Note that any of the four datasets can be used as a reference for this purpose. Among the MW UTH datasets, the UMIAMI dataset has the lengthiest time period of AMSU-B and MHS to allow for the longest MW comparison with others, and it is used as the relative reference in the calculation. Figure 1d shows that the anomaly values are mostly within ±0.5% UTH of each other, with the exceptions during El Niño events when the anomaly differences can be larger. Chung et al. (2016) analyzed the relative differences among the brightness temperatures of the channels sensing upper tropospheric humidity from HIRS, AMSU-B/MHS, and AIRS. The brightness temperature differences between the HIRS and AMSU-B/MHS were mostly within ±0.2 K.

During an El Niño event (such as the 2015-16 and 2009-10 events as displayed in Figure 1e) the infrared dataset tends to have a smaller value of averaged UTH compared to microwave UTH values, and the opposite occurs during a La Niña event (such as the 2010-11 and 2007-08 events). This indicates that the infrared clear-sky dataset may not fully capture the increase of water vapor during El Niño events due to the exclusion of very humid pixels associated with clouds, and tends to have better sampling of the dry regions. Figure 1d also shows that the tropical mean UTH has a larger moistening trend in CMSAF than the other datasets. Allan et al. (2022) presented tropical (30°S–30°N) ocean and land averaged anomaly time series of ERA5 relative humidity (RH), AIRS RH, HIRS UTH, and MW UTH (Figures 6 and 7 of their study). The HIRS and MW UTH are the NCEI and UMIAMI UTH datasets analyzed in the present study, and the features of these two datasets are similar to the NCEI and UMIAMI UTH time series in Figure 1b.

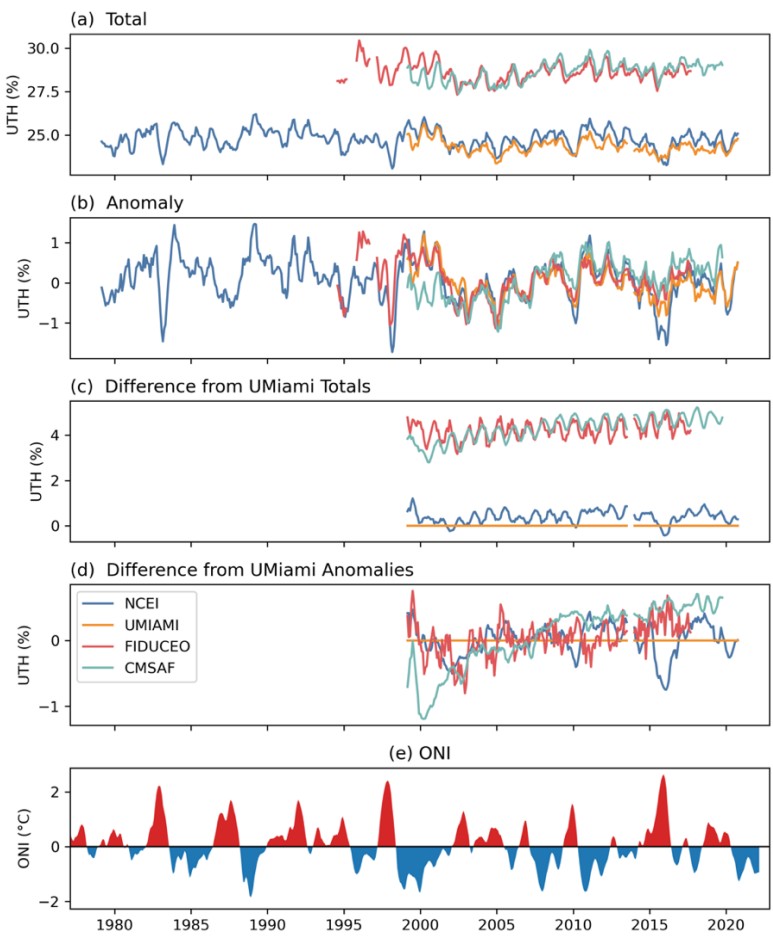

**Figure 1: Time series of UTH (%) averaged over 20°S–20°N for (a) the averaged values of UTH, (b) the corresponding anomalies relative to the 2000-2015 climatology, (c) the differences of UTH values relative to the values of UMIAMI, and (d) the differences of anomaly values relative to those of UMIAMI. A five-month moving average is applied to the UTH time series to reduce short-term fluctuations. Panel (e) shows the time series of ONI.**

During major El Niño events, tropical water vapor fields exhibit distinct characteristics, and the enhanced signals facilitate the comparison of datasets. Figure 2 shows the time series of UTH over the Niño 4 region (equatorial central Pacific 5°S–5°N, 160°E-150°W). Figure 2a shows that the interannual variability of UTH is much larger compared to tropical mean values in Figure 1a, but similar differences between datasets remain. The UTH values of the CMSAF and FIDUCEO datasets are generally larger than the values of NCEI and UMIAMI datasets by approximately 5% UTH on average (Figure 2c). In the anomaly plots (Figure 2b), all datasets depict consistent inter-annual variations. In Figure 2d, the infrared UTH again shows smaller values compared to microwave UTH values during El Niño events and larger values during La Niña events, similar to the features displayed in the tropical average time series in Figure 1d. Though a moistening trend is shown

in the CMSAF UTH time series in Figures 1c and 1d where the tropical average is taken, the moistening trend is not as apparent for the Niño 4 region as displayed in Figures 2c and 2d.

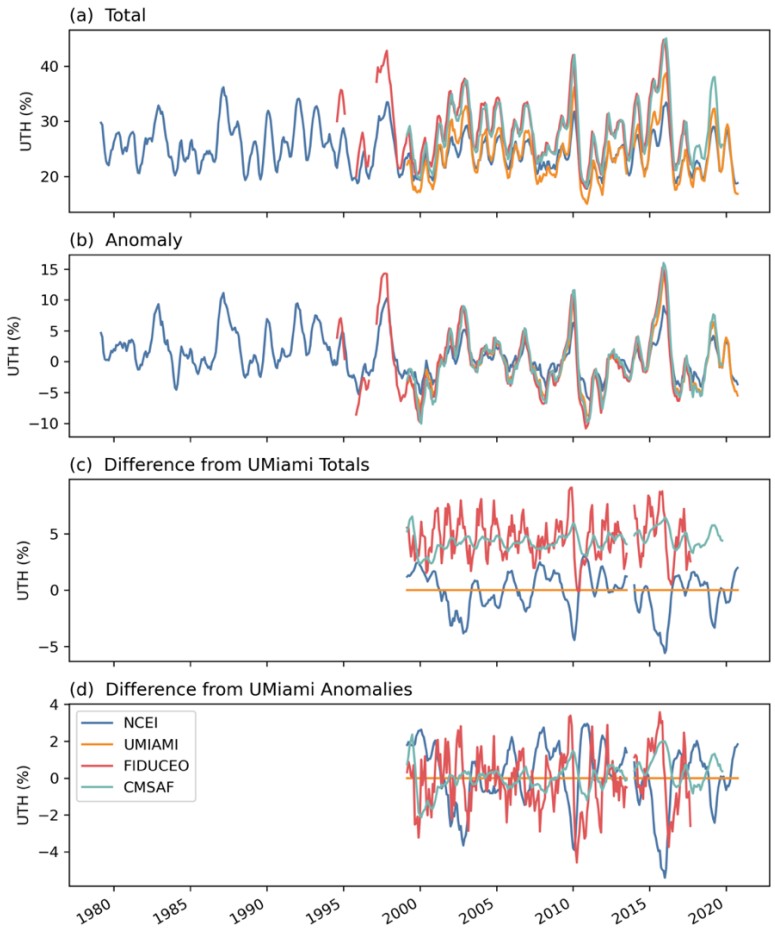

 **Figure 2: Time series of UTH (%) averaged over the Niño 4 region for (a) the averaged values of UTH, (b) the corresponding anomalies relative to the 2000-2015 climatology, (c) the differences of UTH values relative to the values of UMIAMI, and (d) the differences of anomaly values relative to those of UMIAMI. A five-month moving average is applied to the UTH time series to reduce short-term fluctuations.**

It is interesting to observe that between Figures 1b and 2b the phases of the variations are mostly opposite. During the major El Niño events (for example, 1982–83, 1997–98 and 2015–16), the tropical averaged time series exhibited large negative values of anomalies (Figure 1b), while at the same time, large positive anomalies occurred in the Niño 4 region (Figure 2b). An earlier study (Shi et al., 2018) showed that unlike UTH, the total column water vapor (TCWV) in the tropical average exhibited large positive anomalies during El Niño events, having the same phase as the Niño 4 region UTH time series. The

TCWV is largely weighted by water vapor in the lower troposphere. During an El Niño event, there are larger areas of water

vapor increase in the lower atmosphere as reflected in the TCWV field, compared to the UTH field. The enhanced deep convection provides a conduit to transport more water vapor to the atmosphere. However, the increased water vapor in the upper atmosphere is confined to relatively small areas. The study of Lim et al. (2017) showed that during a major El Niño the rising motion of the Hadley circulation is dominant within 10°S–0°. The branch of sinking motion in the subtropics (15°–25°N) is well organized stretching from the surface to the upper troposphere. In the upper troposphere, large positive anomalies of total cloud fraction are formed over 10°S-5°N, and negative cloud anomalies occurred over the subtropics. Beyond the constrained positive UTH anomalies around the equator, the water vapor in the upper troposphere is suppressed in large areas outside the Niño 4 region, which causes large area of negative UTH anomalies, consistent with the sinking motion of the Hadley branch. When a tropical average is taken, the larger areas of negative anomalies over-compensate for the smaller areas of positive anomalies, and result in mean negative anomalies during El Niño events. As the Niño 4 region is the center of enhanced deep convection during El Niño events, the phase of UTH is consistent with that of the water vapor in the lower atmosphere, and consistent with the phase of sea surface temperature during El Niño events as shown in Figure 1e and as described in, e.g., Trenberth (1997), Mcphaden (1999), Wolter and Timlin (2011), Lim et al. (2017), and Santoso et al. (2017).

We use the longitude–time Hovmöllers to examine spatio-temporal variability of UTH over the deep tropics. Figure 3 shows longitude–time evolutions of monthly UTH anomalies around the equator, averaged between 5°S and 5°N for the four datasets. During the past 40 years, the most significant three El Niño events occurred in 1982–83, 1997–98 and 2015–16 according to ONI shown in Figure 1e. During these events the UTH field is marked by increased anomalies in the central-eastern and corresponding decreased UTH in the western equatorial Pacific. All three events can be clearly identified in the NCEI time series, which has the longest temporal coverage.

The 1997-98 and 2015-16 events are also clearly displayed in the FIDUCEO time series. However, the sparsity of the SSM/T-2 data before 1998 can be seen in the noisier appearance of the anomalies during that period. Nonetheless, both the NCEI and FIDUCEO datasets show that the 1997-98 event was marked with higher anomaly values and extended further east in the Pacific in terms of large positive UTH anomalies compared to the 2015-16 El Niño. The Multivariate El Niño/Southern Oscillation Index (MEI) indicates similar differences in the strength of these El Niño events. In addition to the commonly used sea surface temperature (SST) anomalies, the MEI also incorporates surface air temperature, sea-level pressure, zonal and meridional components of the surface wind, and total cloudiness fraction of the sky (Wolter and Timlin, 2011). The Multivariate ENSO Index Version 2 (MEI.v2) values (available at https://psl.noaa.gov/enso/mei/#data; accessed June 3, 2022) show that MEI reached as high as 2.5, and remained at or above 2.0 for 12 consecutive months during the 1997-98 El Niño event. During the 2015-16 El Niño, the MEI was as high as 2.2, and remained above 2.0 for only two months. The UMIAMI and CMSAF UTH time series both started in late 1998, and they have similar patterns in the Hovmöller analysis, both distinctively showing the 2015-2016 El Niño event.

Allan et al. (2022) examined changes in the anomaly characteristics in the zonal mean of AMIP 300–500 hPa RH, ERA5 300–500 hPa RH, and HIRS UTH (their Fig. 8a-c). Both the AMIP and HIRS time series showed a detectable decreasing

trend in UTH 30°S-60°S, and all three datasets showed decreasing amplitudes of anomalies after 2000. More specifically, AMIP and HIRS showed smaller positive anomalies while ERA5 exhibits smaller negative anomalies. The FIDUCEO MW UTH in Figure 3c of our study also shows subtly stronger UTH amplitudes before 2000, albeit with only a few years of data available. These changes after 2000 seem to be coincident with the decrease in the strength of El Niño events after 2000 as depicted by the MEI.v2, though such changes are not displayed in SST-only Niño indices such as the ONI.

During the common period when data are available from all four datasets, the most significant La Niña event occurred in 2010-2011, in which the MEI.v2 value reached -2.4. The UTH field was marked by decreased UTH in 120°E-160°W and increased UTH in 80°E-120°E. The event can be seen from all UTH datasets. In general, the equatorial UTH anomalies in the infrared measurements are relatively weaker than those in the microwave measurements. The definition used to compute the HIRS UTH may be the primary factor for the smaller magnitude. The averaging of pixel-level brightness temperatures to

the grids first before the UTH is computed may further smooth out the largest anomalies (both positive and negative).

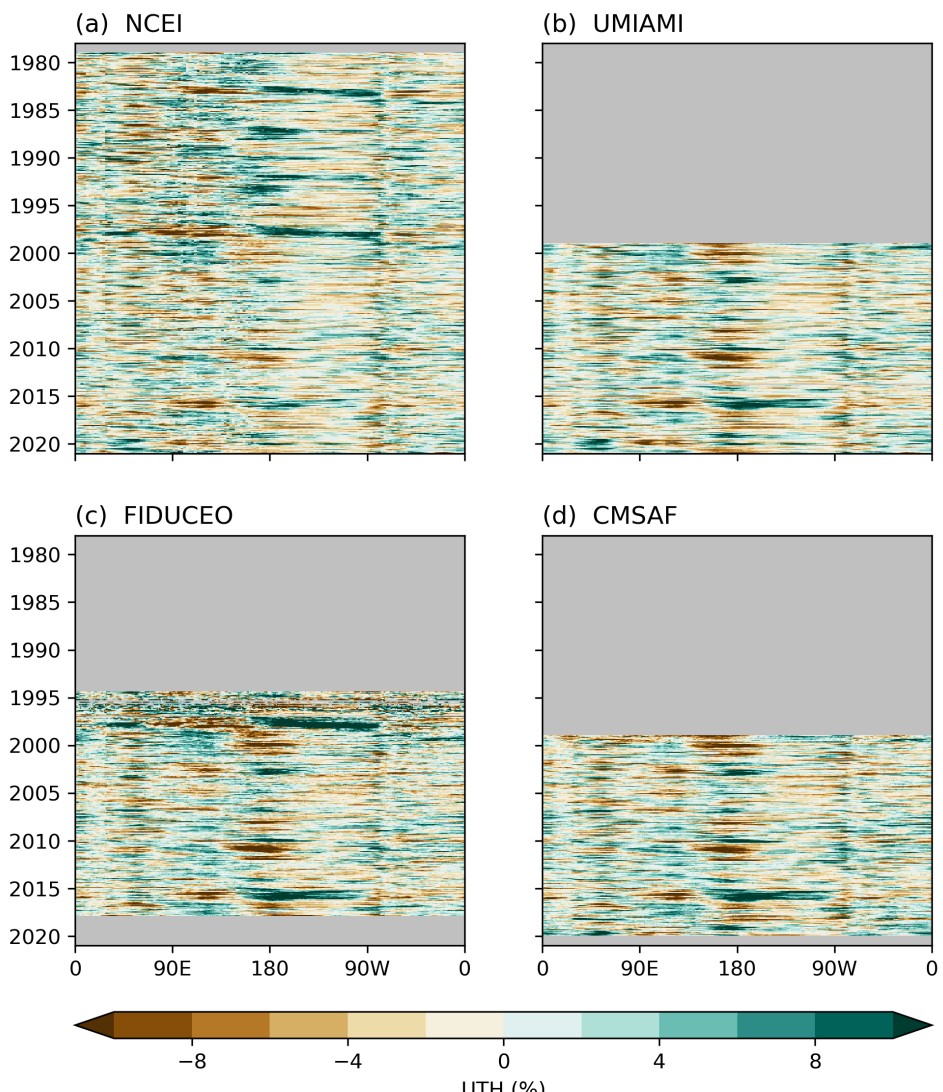

**Figure 3: Time versus longitude section of UTH monthly anomaly. The analysis is based on an average of data between 5°N and 5°S.**

To quantify the changing proportion of dry and humid regions derived from the different datasets, we calculate the percentage of grids with anomaly values greater or less than several selected values over 20°S-20°N (Figure 4). The anomalies are relative to each of the grid points and deseasonalized before the percentages are calculated. Grids with UTH anomaly values > 5% represent very humid anomalies while those < -5% represent very dry anomalies. Among the MW

datasets, the SSM/T-2 derived UTH in the FIDUCEO series has the highest proportion of very humid anomalies. For the AMSU-B/MHS series, FIDUCEO dataset generally has 2-4% more very humid anomalies than those of the UMIAMI

dataset. The gridding of UTH after the pixel-level brightness temperature values are averaged in the UMIAMI dataset may have smoothed out some of the most humid measurements. The CMSAF UTH has fewer dry anomalies before 2005 than the other datasets, but it has the largest proportion of very humid anomalies in recent years. The infrared dataset has the smallest proportion of humid anomalies compared to the MW datasets at both levels (> 5% and > 1%) due to the exclusion of cloudy pixels.

HIRS UTH also generally has the smallest proportion of the driest anomalies (< -5%), but the ratios are often close to those of the UMIAMI dataset. Interestingly, when the majority of the negative anomalies are examined (UTH anomalies < -1% in Figure 4b), the HIRS dataset frequently has the largest ratios of the dry anomalies. This phenomenon is particularly significant during both major El Niño and La Niña events. For example, during the 2015-16 El Niño, the ratios of UTH anomalies < -1% are approximately 51% for HIRS, 47% for UMIAMI, 46% for FIDUCEO, and 45% for CMSAF dataset. In other words, the HIRS data identifies more dry anomalies than the MW datasets, though the magnitude of the driest HIRS UTH does not usually reach as large values as those of the MW UTH likely due to the definition of the UTH formula used. Overall, the FIDUCEO dataset has the largest amplitude of the ratios for both the most humid and driest measurements.

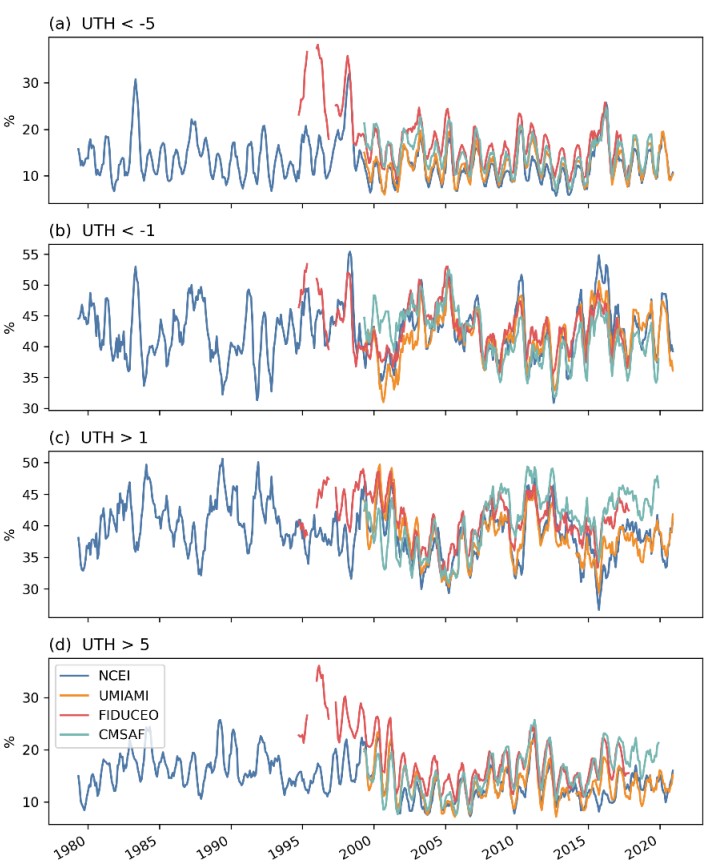

**Figure 4: Time series of the ratios of grids over 20°S-20°N with anomaly values less than -5% and -1%, and greater than 1% and 5%.**

### 3.2 Spatial anomalies during major El Niño and La Niña events

During the common period of the four datasets, the most significant El Niño and La Niña events occurred in 2015-2016 and 2010-2011, respectively. The spatial patterns of UTH anomalies for 60°S-60°N during the peak six months of the 2015-16 El Niño event are shown in Figure 5. The anomalies of several environmental variables, including data from the Global Precipitation Climatology Project (GPCP), NOAA Extended Reconstructed SST V5 (ERSSTv5), and modeled 200 hPa velocity potential, for the same peak six-month period of the 2015-16 El Niño are displayed in Figure 6 to show the large-scale atmospheric circulation and SST fields. The GPCP data are generated by combining satellite retrieval and in situ precipitation into a final merged gridded product (Adler et al., 2003). The ERSSTv5 dataset is derived from the International Comprehensive Ocean–Atmosphere Dataset (ICOADS) and is available at gridded monthly global coverage (Huang et al., 2017). Velocity potential anomalies at 200 hPa are taken from the Climate Forecast System Reanalysis (CFSR) (Saha et al., 2010) for 2000–2010 and the related Climate Forecast System v2 (CFSv2) operational analyses (Saha et al., 2014) for 2011–2016.

Similar to that discussed in Shi et al. (2018), during the 2015-16 El Niño event, UTH developed strong positive anomalies over the equatorial central Pacific, extending to the eastern Pacific in 5°-10°N. The enhanced El Niño convection drove compensating subsidence and thus negative UTH anomalies surrounding the positive anomalies. The positive SST anomalies were centered along the equatorial central-eastern Pacific (Figure 6b). Anomalous divergence developed over the warmed SST and was balanced by the anomalous convergence over the western Pacific and the Indian Ocean (Figure 6c). The pattern of positive anomalies of UTH above the Niño 4 region and along 5°-10°N in the eastern Pacific highly resembles the pattern of the positive precipitation anomalies (as shown in Figure 6a), indicating the strong linkage between the two variables. Similar pattern of precipitation during the 2015-16 El Niño was also shown in the study of Santoso et al. (2017).

Overall, the area of the strong positive UTH anomalies over the equatorial central Pacific is smaller than the surrounding areas of strong negative anomalies in the tropics. Taking an example of the FIDUCEO UTH dataset, there are approximately 34% of grids in the tropical domain 20°S-20°N that have UTH anomalies greater than 1%, compared to more than 49% of grids having UTH anomalies less than -1% at the peak of the 2015-16 El Niño as shown in Figure 4. The other three datasets also show larger portions of dry grids than humid grids during the event. When a tropical domain average of anomalies is taken, it results in a negative anomaly during an El Niño event as shown in Figure 1. In the NCEI HIRS UTH panel, the magnitudes of both positive anomalies along the central-eastern equatorial Pacific and the negative anomalies in the western Pacific appear smaller than those in the other three microwave UTH panels, consistent with what is seen in the Hovmöller analysis discussed earlier. However, over the tropical domain, the HIRS data have larger proportions of dry areas in the subtropics during El Niño events (resulting in larger overall dry area ratios shown in Figure 4b), leading to deeper dips of UTH during El Niño events displayed in Figure 1b.

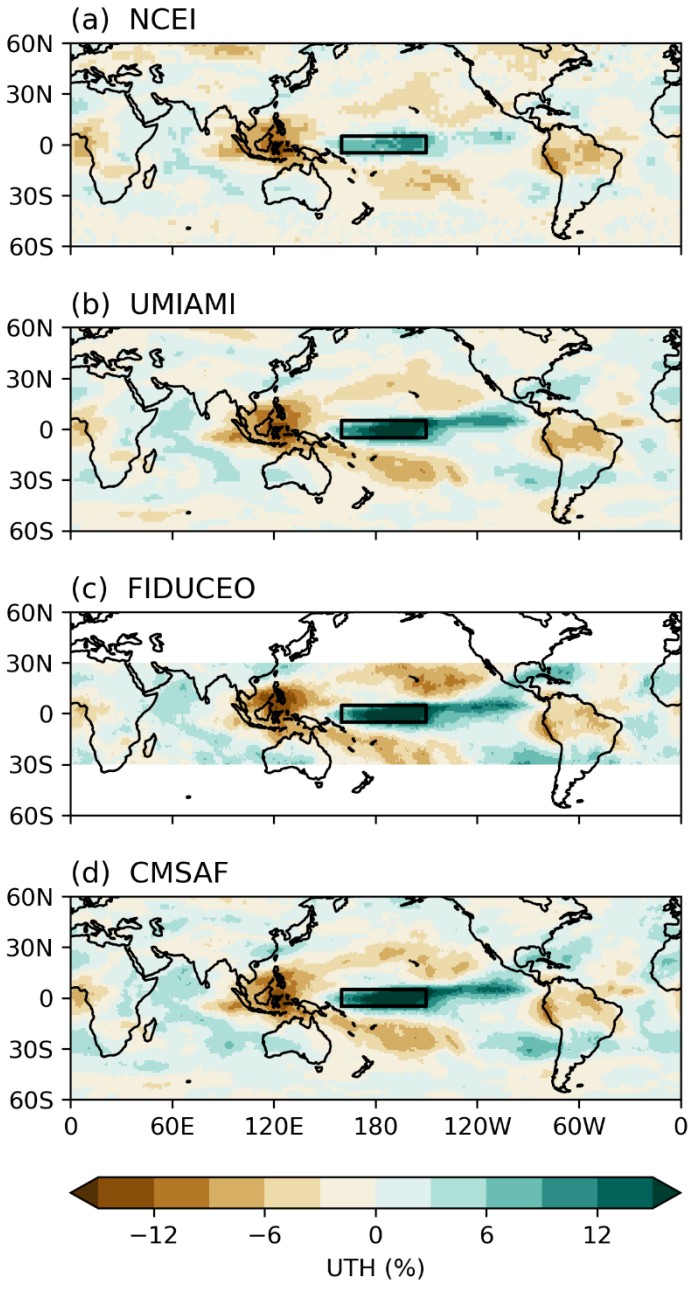

**Figure 5: Anomalies of UTH during the peak six months of the 2015-16 El Niño event. The box shows the Niño 4 domain in the central Pacific (5°S–5°N, 160°E–150°W).**

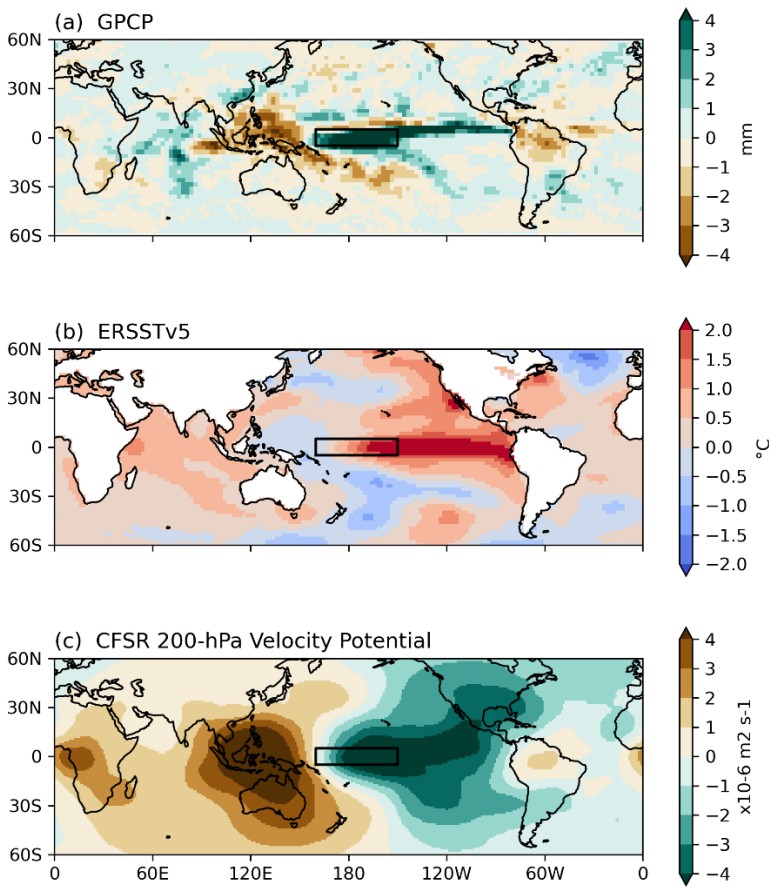

**Figure 6: Anomalies of GPCP precipitation, ERSSTv5 SST, and CFSR 200-hPa velocity potential during the peak six months of the 2015-16 El Niño event. The box shows the Niño 4 domain in the central Pacific (5°S–5°N, 160°E–150°W).**

To further assess the consistency of UTH datasets with several environmental variables, histograms of UTH anomalies vs. anomalies of GPCP precipitation, ERSSTv5 SST, and CFSR 200-hPa velocity potential during the peak six months of the 2015-16 El Niño are presented in Figures 7-9. The correlations between the anomalies of UTH and those of the three variables are also calculated and the correlation values are labelled (as Corr) at the top of each panel. Among the three variables, precipitation has the highest correlations with UTH (Figure 7), while SST has the lowest (Figure 8). Both precipitation and velocity potential are proxies for vertical motion, so they are more directly tied to wet/dry UTH than the SST surface forcing. The increases of SST during El Niño events usually occur most significantly in the equatorial eastern-central Pacific, while the increases of both UTH and precipitation are more confined over the equatorial central Pacific. The UTH and precipitation fields both have a more balanced dipole between the central and western equatorial Pacific during a major El Niño, while the decrease of SST in the western equatorial Pacific does not match the strength of positive anomalies in the central-eastern equatorial Pacific. These patterns lead to overall higher correlations between UTH and precipitation

than those between UTH and SST. The correlation values also illustrate that an SST-only ENSO index may not be as good of an indicator for the strength of UTH compared to an index that includes other environmental variables such as the MEI.

Among the UTH datasets, the MW data have higher correlations with the three environmental variables. The HIRS UTH correlation values are about 0.1 lower, primarily due to the lack of very humid anomalies in the infrared dataset. The histograms show that for all UTH datasets, the highest densities of anomalies are consistently centered around zero. The

340 density of HIRS positive anomalies decreases rapidly beyond 5%, in line with the lowest ratio of large HIRS UTH shown in Figure 4d.

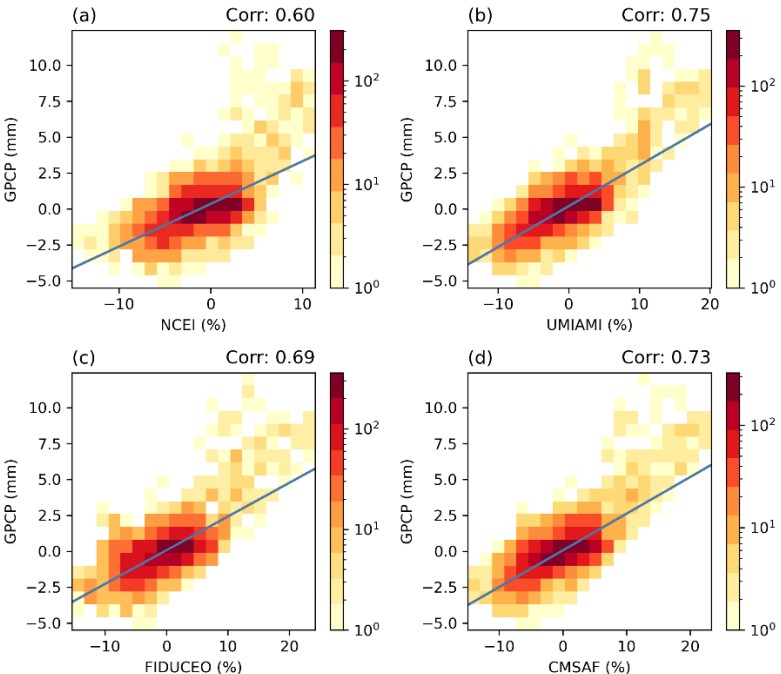

**Figure 7: Histograms of UTH anomalies of the four datasets vs. anomalies of GPCP precipitation during the peak six months of**
345 **the 2015-16 El Niño. The blue line represents the linear regression line. The correlations between UTH anomalies and GPCP precipitation anomalies are labelled at the top of the panels.**

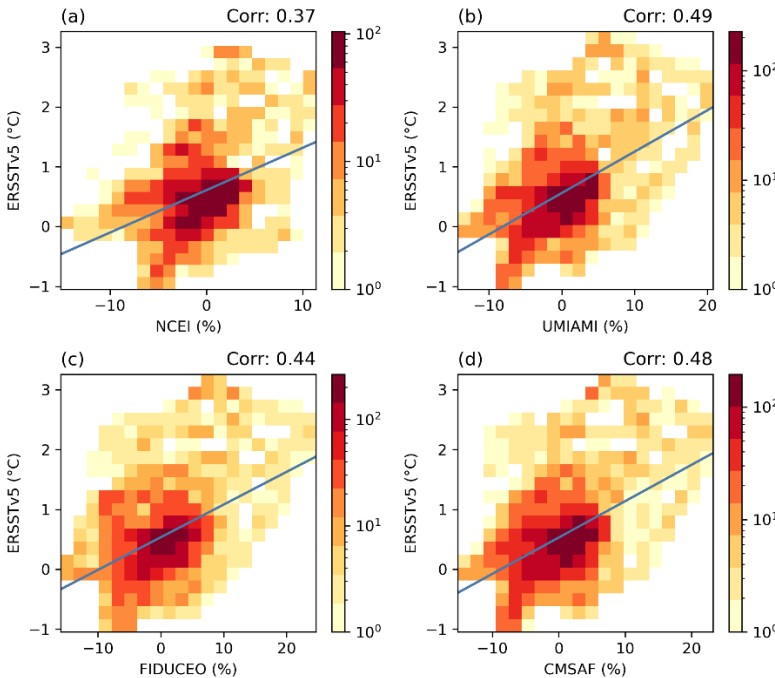

Figure 8: Similar to Figure 7 except for UTH anomalies vs. ERSSTv5 anomalies.

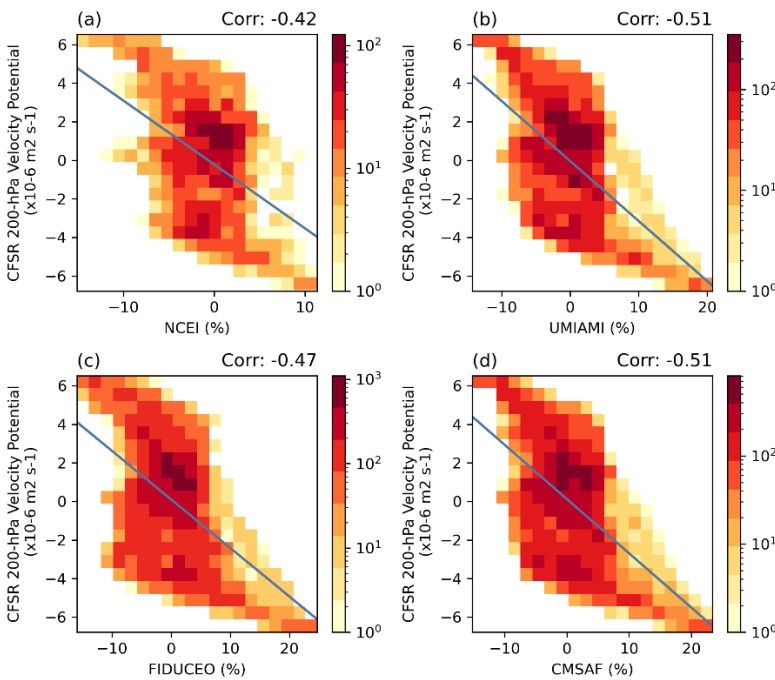

Figure 9: Similar to Figure 7 except for UTH anomalies vs. anomalies of CFSR 200-hPa velocity potential.

Figure 10 shows the UTH anomaly fields averaged over six months near the peak of the La Niña in 2010-11, and Figure 11 displays the anomalies of GPCP, ERSSTv5, and CFSR 200 hPa velocity potential data for the same time period. During a La Niña event, the central Pacific and Indonesia exhibited mostly opposite signs of anomalies for UTH, SST, precipitation, and 200 hPa velocity potential compared to the El Niño patterns depicted in Figures 5 and 6, except that the negative anomalies of the 200 hPa velocity potential were more confined to the center over Indonesia and Australia. La Niña events tend to lead to significant increases of UTH over Indonesia and the equatorial eastern Indian ocean and over Pacific subtropics, and decreases of UTH over the Niño 4 region. Slightly positive UTH anomalies may be found in the equatorial eastern Pacific during a La Niña event. Similar patterns of tropical features are shown in all four datasets, although the magnitudes are again smaller in the infrared UTH (Figure 10a).

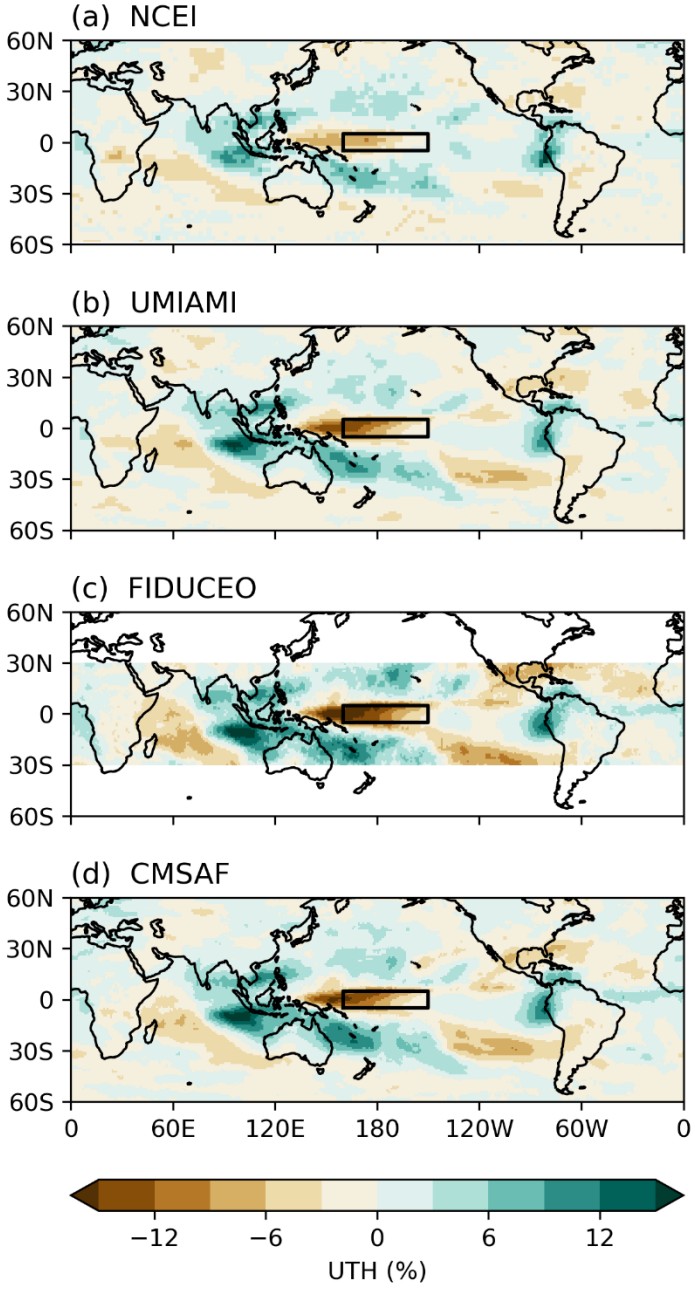

**Figure 10: Anomalies of UTH during the peak six months of the 2010-11 La Niña event. The box shows the Niño 4 domain in the central Pacific (5°S–5°N, 160°E–150°W).**

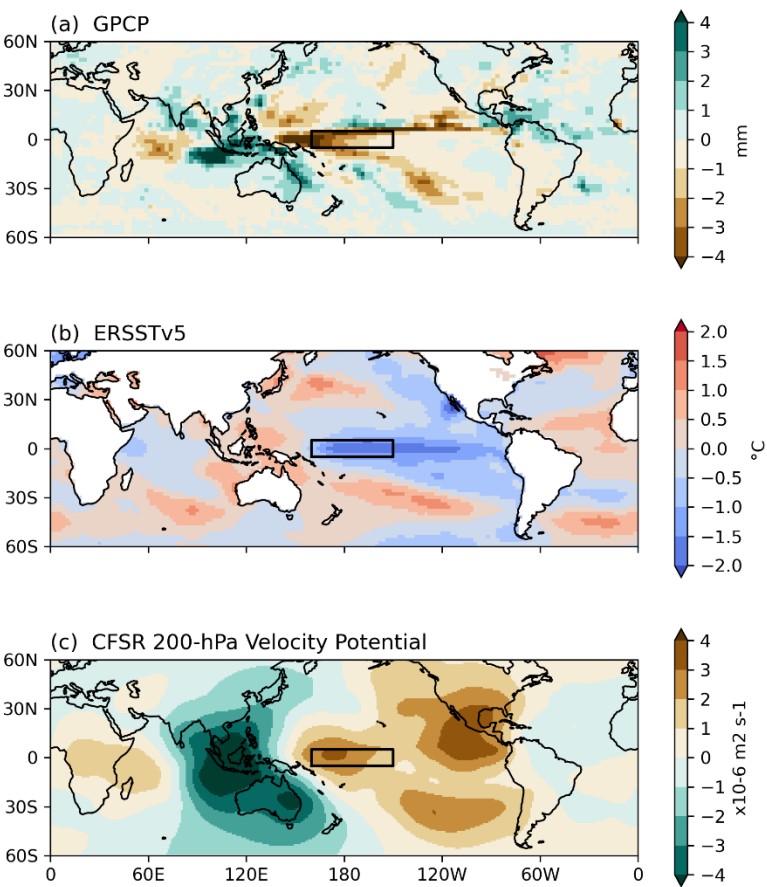

**Figure 11: Anomalies of GPCP precipitation, ERSSTv5 SST, and CFSR 200-hPa velocity potential during the peak six months of the 2010-11 La Niña event. The box shows the Niño 4 domain in the central Pacific (5°S–5°N, 160°E–150°W).**

### 3.3 UTH changes during the common period of the datasets

The common period when all four UTH datasets have data spans from 1999 to 2017. To analyze UTH changes of each dataset during the common period, we use the linear trend method to calculate the change rate of each grid, and the results are displayed in Figure 12. In this study, the linear trend method is employed to show the change rates during a relatively short common period as a way to examine dataset consistency, and the results should not be interpreted as long-term trends. The La Niña event in 1998-2000 at the beginning of the common period and the strong El Niño event in 2015-16 near the end of the common period can significantly impact the resulting trend values. The Mann-Kendall test is used to test the significance of the trends at each grid. The trends appear to be significant at 0.95 only in a few small places, mainly sparsely spotted along subtropical Pacific belts of negative change rates (not plotted in Figure 12), indicating that the time series is

too short for a meaningful trend study for the majority of areas. In the present study, the trend results are only used as a consistency evaluation of the datasets.

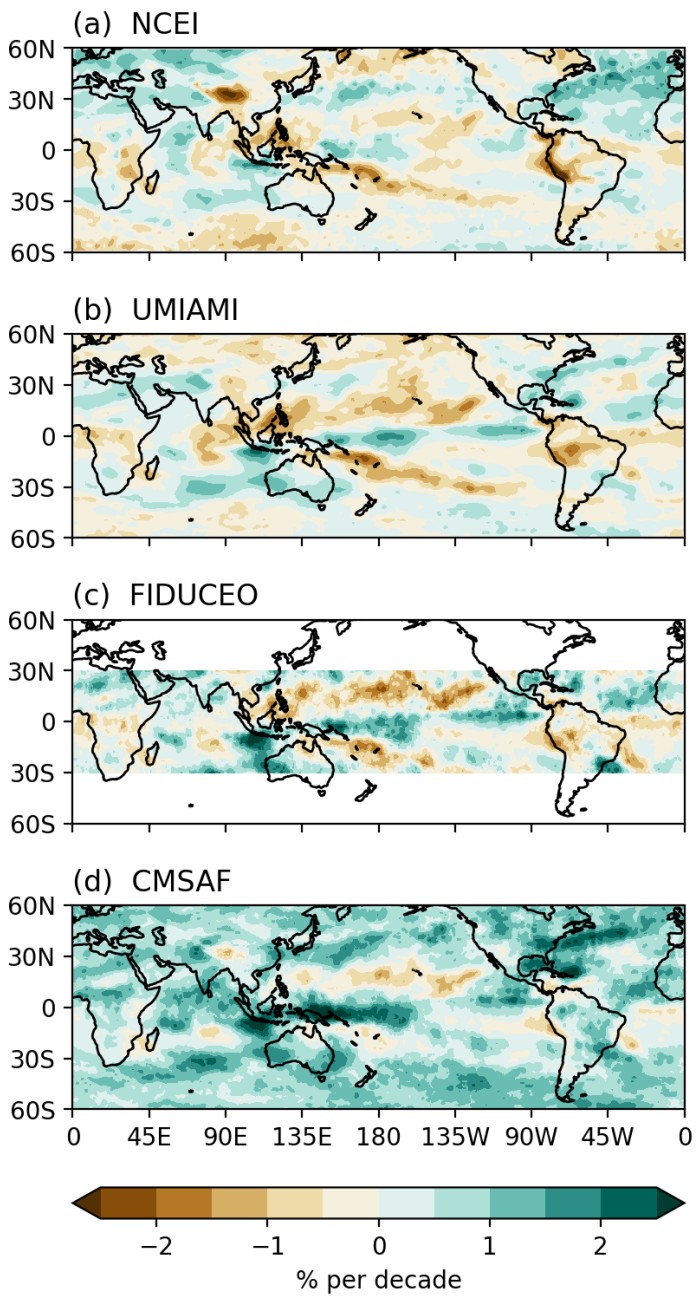

Figure 12: Change rates of the four UTH datasets during the common period 1999 to 2017.

General consistency of the change patterns in the tropics is found among the four datasets. They all show increased UTH over the Niño 4 region (5°S–5°N, 160°E-150°W) and over the eastern Pacific near 5°N-10°N, and a decrease of UTH over Peru and surrounding areas. Decreased UTH values along both the northern and southern Pacific subtropics are seen in all datasets. The change rate patterns over the tropical and subtropical Pacific follow the 2015-16 El Niño UTH patterns (as shown in Figure 5) to some extent, indicating the influence of the El Niño signals on the change rate calculation. Over the Indian Ocean, decreased UTH centered over the equatorial central Indian Ocean is surrounded with increased UTH in most datasets, except that the center of decreasing rates is confined to a smaller area around 15°S for the CMSAF UTH. The change rates (both positive and negative) in the NCEI HIRS dataset (Figure 12a) are generally smaller than those in microwave datasets. The largest change rates are found in the CMSAF image, with positive changes covering most of the areas, consistent with the trend in Figure 1d. An earlier study (Lang et al., 2020) plotted the time series of individual satellite's UTH from NOAA-15 to Metop-B for both FIDUCEO and CMSAF datasets (Figure 6 in that article). Their figure 6b showed that offsets between the UTH time series from consecutive satellite missions in the CMSAF record tend to be positive over time. When all the satellites are merged into one time series this may lead to a positive trend.

The three datasets that have mid-latitude coverage (Figure 12a, b, and d) exhibit negative change rates over the Tibetan Plateau. This may not necessarily indicate a decrease in water vapor, though. Over high elevations (similar to over high latitudes) there are contributions of the surface temperature to the radiances measured by satellite UTH sounders. A decrease in calculated UTH values over a high elevation can be caused by either a decrease in water vapor or an increase in the surface temperature. The clear-sky measurement excludes some high humidity data due to removal of cloudy pixels compared to MW datasets. The Jacobian of less-humid data has a lower peak in the atmosphere, and the lower tail of the Jacobian profile is closer to the surface (e.g., see Figure 1 in Brogniez et al. (2006)). Over a high elevation, increasing surface effect can be included in the observation radiances. A warming at the surface may contribute more to an infrared dataset due to larger portion of less-humid data. Over the mid-latitude Pacific, both NCEI and UMIAMI data show negative change rates in 45-60N, while the CMSAF dataset shows positive change rates. Over the southern hemisphere mid-latitude, The CMSAF dataset displays increased humidity, while both positive and negative change rates are found in the NCEI and UMIAMI datasets.

## 4 Conclusions

In this study we assess the consistency of four UTH datasets derived from both microwave and infrared sounders of polar-orbiting satellites as part of the GEWEX water vapor assessment activities. These include measurements from the $183.31 \pm 1$ GHz channel on SSM/T-2, AMSU-B, and MHS and HIRS channel 12 (calibrated to 6.7 µm). The main conclusions are:

1. The four datasets are consistent in tropical spatial patterns and in interannual variability. Large positive anomalies peaked over the Niño 4 region during El Niño events in the same phase with the increase of sea surface temperature

as expected. At the same time, opposite phases of anomalies were obtained in the averaged tropical anomalies because the compensating drying areas of dissipation are larger than the relatively confined moistening area above deep convection. All four datasets exhibit such similar temporal variability.

2. The infrared UTH dataset exhibits the largest proportions of dry areas at the peak of El Niño and La Niña events (more than 4% larger ratio of dry areas compared to those of MW datasets). The MW datasets have larger proportion of humid measurements during El Niño events, while during a major La Niña such as the 2010-11 event, the ratios of humid areas are close to each other among three UTH dataset (differences less than 1%), except the CMSAF dataset which overall has larger humid areas.

3. Through the common period of 1999 to 2017, differences are observed in the changing rates of the datasets. Wider spread of UTH moistening is observed in the CMSAF datasets.

4. The four datasets show that during a major El Niño event, there are significant increases of UTH over a narrow belt of the equatorial central Pacific consistent with the positive anomalies of the precipitation pattern, though typically the positive anomalies of SST cover a larger latitude span and are more prominent in the eastern Pacific. Negative anomalies develop over the weakened ascending branch of the Pacific Walker circulation in the western Pacific and eastern Indian ocean where there is a positive anomaly of the 200 hPa velocity potential, and over the enhanced descending branches of the local Hadley circulation along the Pacific subtropics.

5. During a major El Niño, the spatial correlations between UTH and SST are not high, with the correlation values in the range of 0.37 to 0.49. In the meantime, the spatial correlations between UTH and precipitation are higher, ranging in 0.60 to 0.75. The infrared dataset has lower correlation values (about 0.1 smaller) with SST, precipitation, and 200 hPa velocity potential compared to those for the MW UTH datasets due to the lack of very humid data in the infrared dataset.

6. Though there are apparent and expected differences in the values of total UTH due to differences in the definition and in the gridding procedure, the tropical-averaged anomalies of the datasets are close to each other (mostly within ±0.5% over tropical domain average), and more importantly the phases of the time series are generally consistent for variability studies.

7. The infrared and MW UTH datasets have their own strengths and weakness. The HIRS dataset has the longest, over 43 years of observations so far, for long-term studies, and its variability, temporal phases, and spatial patterns are generally consistent with MW observations. However, being a clear-sky dataset, it does not capture the most humid regions. The MW datasets have a shorter time series, but they retain almost all-sky data, removing only the precipitating pixels, thus have a better sampling for a full spectrum of UTH especially for very humid data.

**Data availability:** The CMSAF UTH dataset can be downloaded from https://wui.cmsaf.eu/safira/action/viewDoiDetails?acronym=UTH_V001. The FIDUCEO UTH data are obtained from

ftp://ftp-projects.cen.uni-hamburg.de/arts/fiduceo/. The NCEI HIRS channel 12 brightness temperature data are available at https://www.ncei.noaa.gov/data/hirs-brightness-temperature/access/.

**Author contribution**: The research was designed by L.S. and C.J.S.. C.J.S. performed data analyses. L.S. wrote the paper with input from co-authors. L.S., C.J.S., V.J. and E.-S.C were involved in discussions on datasets and analysis results at various stages of the study. T.L., S.A.B., and B.J.S. provided valuable comments. All of the authors reviewed the manuscript and provided input.

**Competing interests:** The authors declare that they have no conflict of interest.

**Acknowledgements:** This study is part of the GEWEX water vapor assessment (G-VAP) organized by the GEWEX Data and Assessments Panel (GDAP). Carl Schreck was supported by NOAA through the Cooperative Institute for Satellite Earth System Studies under Cooperative Agreement NA19NES4320002. We thank Jessica Matthews and Associate Editor Shu-Peng Ho for reviewing the preprint, and thank Reviewer Richard Allan and two anonymous reviewers for constructive comments and suggestions.

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
