# Peer review of "Assessing the consistency of satellite derived upper tropospheric humidity measurements"

_Atmospheric Measurement Techniques, 2022_

## Author Response (AR1)

We thank reviewer Richard Allan and two anonymous reviewers for taking the time to review our manuscript, engaging in collegial discussion, and providing constructive comments and suggestions. These have been helpful in improving the manuscript. The manuscript is revised as described below. For clarity, reviewer's original comments are included in black, our responses are written in blue, and the revision in the updated manuscript is marked with green.

**Author response to Reviewer #1, Richard Allan**

An assessment of four satellite upper tropospheric humidity (UTH) data sets are made with a focus on interannual variability and the effect of El Nino Southern Oscillation (ENSO) on spatial anomalies. Although the spatial anomalies associated with ENSO variability are well known, the strength of this article is in comparing 4 contrasting datasets. In general, users of these or similar datasets will probably wish to know the best dataset for their application so any statements on the relative quality of the datasets and any issues such as inhomogeneities or artificial drifts will add value to the paper. In particular, are there any spurious jumps or trends such as HIRS before and after 2000 or jumps in the CMSAF record? It would also be useful to comment on whether there are systematic biases for any particular meteorological regimes (e.g. convective, anvil, descent, etc).

Thank you for the comments. To assist users of similar datasets, additional statements on the relative strength and weakness of infrared and microwave (MW) UTH datasets are added to the conclusions. Regarding HIRS data, we noted that after 2000 the magnitudes of anomalies during El Nino events appear slightly smaller than those before 2000, but such pattern is also inferred by the FIDUCEO MW UTH dataset (albeit with a short data length before 2000). The Multivariate El Niño/Southern Oscillation Index (MEI), which incorporates surface air temperature, sea-level pressure, zonal and meridional components of the surface wind, and total cloudiness fraction of the sky together with the SST field, also shows noticeable stronger El Nino events in the 1980s and 1990s compared to those after 2000, indicating a possibility of decadal or multi-decadal change of some meteorological variables. The jumps between satellites in the CMSAF UTH dataset were discussed in the Lang et al. (2020) paper, and these were described but the analysis was not repeated in the current article. The UTH datasets have been mostly used in large-scale interannual variability studies, and thus the current consistency assessment is focused on large-scale interannual regimes such as ENSO, rather than on mesoscale meteorological regimes.

**Specific comments:**

Abstract: some more quantitative statements would be useful in assessing the magnitude of differences.

Some quantitative statements are added to the abstract in the revision:

During a major El Niño event, UTH had higher correlations with the coincident precipitation (0.60 – 0.75) and with 200 hPa velocity potential (-0.42 - -0.64) than with SST (0.37 - 0.49). Due to differences in retrieval definitions and gridding procedures, there can be a difference of 3-5% UTH between datasets on average, and more significant anomaly values are usually observed in the microwave UTH data. Nevertheless, the tropical-averaged anomalies of the datasets are close to each other with their differences

being mostly less than 0.5% over tropical domain average, and more importantly the phases of the time series are generally consistent for variability studies.

L45 - application for model evaluation and processes understanding studies could be highlighted

Some references are added in the Introduction to cite studies that use UTH measurements to evaluate climate models for large-scale atmospheric process studies:

The UTH datasets facilitated the evaluation of climate models and contributed to a better understanding of large-scale atmospheric processes (Allan et al., 2003; Soden et al., 2005; Chung et al., 2016; Allan et al., 2022; John et al., 2021).

L131 - presumably non-linearities in the UTH calculation affect the computation of UMIAMI UTH from gridded data and this will affect the absolute magnitude but probably not the anomalies

Agree that the non-linearities in the UTH calculation affect the computation of UMIAMI UTH from gridded data in the absolute magnitude but this process doesn't significantly affect domain-averaged anomaly values. We edited the sentence and now it reads:

Based on a study by John et al. (2006), such different ways of computing UTH can lead to a difference of up to 6% UTH due to the non-linearities in the formula that calculates UTH from brightness temperature values.

Fig.1 - it is expected that HIRS UTH will be lower than microwave estimates since they sample systematically drier, clear-sky scenes. Can the lower UMIAMI values compared to FIDUCEO and CMSAF be explained by the method, in which case why is UMIAMI used as the baseline since it is not computed using swath data?

The lower UMIAMI values can be explained by the gridding method. The purpose of Figures 1c-d and 2c-d is to quantify the relative differences between datasets. UMIAMI UTH has the longest AMSU-B/MHS data, and thus was chosen to be the reference in Figs. 1 and 2. Using a different dataset as a reference would not change the conclusion on relative differences. The following is a plot using FIDUCEO UTH as a reference. The SSM/T-2 UTH in the FIDUCEO time series is not as stable as the AMSU-B/MHS data due to frequent missing data.

[Figure]

To clarify the choice of using UMIAMI UTH as the reference, the following is added to the revision:

To quantify the differences between datasets, the relative differences are calculated. Note that any of the four datasets can be used as a reference for this purpose. Among the MW UTH datasets, the UMIAMI dataset has the lengthiest time period of AMSU-B and MHS to allow for the longest MW comparison with others, and it is used as the relative reference in the calculation. Figure 1d shows that the anomaly values are mostly within ±0.5% UTH of each other, with the exceptions during El Niño events when the anomaly differences can be larger.

L147 and throughout - more quantification of the difference between datasets would be helpful

We think that your comment about L147 refers to the sentence "The values of CMSAF and FIDUCEO UTH are larger than the values of NCEI and UMIAMI UTH." in L146-147. The differences between values of the two groups were discussed and quantified in the following sentence in the same paragraph (L150 of the preprint): "Figure 1c shows that there is a difference of approximately 3-5% UTH between two groups of UTH datasets when a tropical average is taken". Taking your suggestion, we have added quantified difference values between datasets in various places of the revised manuscript.

L153 - please provide quantification of "good agreement"

The "good consistency" in L152 was referred to agreement in interannual variability patterns. The sentence has been edited to clarify the discussion, and one more sentence is added to better describe the patterns:

In spite of this structural discrepancy, the anomaly plot of the UTH in Figure 1b shows consistency in seasonal and interannual variability patterns among the four datasets. All four datasets show major peaks and dips along the time series in the same phases, though there are differences in the magnitudes of the fluctuations.

L155 - how close is "close to each other" in the context of their use?

The closeness of the dataset anomalies was quantified in the sentence that immediately follows (in L155-L156 of the preprint. The sentence is edited to:

Figure 1d shows that the anomaly values are mostly within ±0.5% UTH of each other, with the exceptions during El Niño events when the anomaly differences can be larger.

L157 (and L256) - is the smaller variability in HIRS linked to the fact it is not sampling the full scale of meteorological regimes (e.g. clear-sky only and so sampling less of the tropical deep convective regions)?

We agree. The clear-sky requirement in the HIRS datasets excluded the majority of deep convective regions. We added "as deep convective regions are excluded" to the sentence as:

This indicates that the infrared clear-sky dataset may not fully capture the increase of water vapor during El Niño events due to the exclusion of very humid pixels associated with clouds, and tends to have better sampling of the dry regions.

Section 3.1 - Variability in UTH and RH in a range of datasets are shown in Figures 6-8 of Allan et al. (2022) which could be commented on. Can the shifting of wet regions more over land during La Niña and more over the ocean contribute to changes in the biases since the retrieval over land and ocean may differ subtly? Or does it relate more to the changing proportion of "dry" and "humid" regions that are sampled differently by the different instruments (particularly HIRS)? A metric for proportion of the tropics with UTH>X and UTH<X, where X could be 50% or a suitable mid-range value, would be interesting.

A reference to Allan et al. (2022) is added to the revision:

Allan et al. (2022) presented tropical (30°S–30°N) ocean and land averaged anomaly time series of ERA5 relative humidity (RH), AIRS RH, HIRS UTH, and MW UTH (Figures 6 and 7 of their study). The HIRS and MW UTH are the NCEI and UMIAMI UTH datasets analyzed in the present study, and the features of these two datasets are similar to the NCEI and UMIAMI UTH time series in Figure 1b.

Please also see the discussion added for Fig. 3 below that references on Figure 8 of Allan et al. (2022).

Regarding the wet anomalies during La Niña, our figures show that they are primarily the result from the vast moistened Pacific subtropics.

We incorporate your suggestion of plotting the proportion of wet and dry regions (but using anomalies rather than total values), and include it as the new Figure 4 in the revision. The new figure and discussions are copied below:

[Figure]

**Figure 4: Time series of the ratios of grids over 20°S-20°N with anomaly values less than -5% and -1%, and greater than 1% and 5%.**

To quantify the changing proportion of dry and humid regions derived from the different datasets, we calculate the percentage of grids with anomaly values greater or less than a fixed value over 20°S-20°N (Figure 4). Grids with UTH anomaly values > 5% represent very humid anomalies while those < -5% represent very dry anomalies. Among the MW datasets, the SSM/T-2 derived UTH in the FIDUCEO series has the highest proportion of very humid anomalies. For the AMSU-B/MHS series, FIDUCEO dataset generally has 2-4% more very humid anomalies than that of the UMIAMI dataset. The gridding of UTH after the pixel-level brightness temperature values are averaged in the UMIAMI dataset may have smoothed out some of the most humid measurements. The CMSAF UTH has fewer dry anomalies before 2005 than the other datasets, but it has the largest proportion of very humid anomalies in recent years. The infrared dataset has the smallest proportion of humid anomalies compared to the MW datasets at both levels (> 5% and > 1%) due to the exclusion of cloudy pixels.

HIRS UTH also generally has the smallest proportion of the driest anomalies (< -5%), but the ratios are often close to those of the UMIAMI dataset. Interestingly, when the majority of the negative anomalies are examined (UTH anomalies < -1% in Figure 4b), the HIRS dataset frequently has the largest ratios of

the dry anomalies. This phenomenon is particularly significant during both major El Niño and La Niña events. For example, during the 2015-16 El Niño, the ratios of UTH anomalies < -1% are approximately 51% for HIRS, 47% for UMIAMI, 46% for FIDUCEO, and 45% for CMSAF dataset. In other words, the HIRS data identifies more dry anomalies than the MW datasets, though the magnitude of the driest HIRS UTH does not usually reach as large values as those of the MW UTH likely due to the definition of the UTH formula used. Overall, the FIDUCEO dataset has the largest amplitude of the ratios for both the most humid and driest measurements.

Fig. 2 - panels c and d do not seem to add much to a and b so could be removed since they are barely referred to

We would keep both panels. We intend to show that while between Figures 1b and 2b the phases of the variations are mostly opposite, the signs (positive and negative) of the difference between infrared and MW UTH datasets remain the same between Figures 1c and 2c and between Figures 1d and 2d. Additional description regarding the moistening trend in the CMSAF UTH dataset shown in the two panels is also added to the paragraph:

Though a moistening trend is shown in the CMSAF UTH time series in Figures 1c and 1d where the tropical average is taken, the moistening trend is not as apparent for the Niño 4 region as displayed in Figures 2c and 2d.

Fig. 3 - there seems to be a change in the anomaly characteristics in NCEI after 2000. Does this relate to the unusual series of La Niña events associated with slower global warming in the 2000-2012 period or are there changes in the instrument? Interestingly the anomaly characteristics of the microwave data after 2000 seem more like the pre-2000 NCEI record than the coincident post-2000 period. A similar change in anomaly characteristics seems present in Figure 8c of Allan et al. (2022) https://doi.org/10.1029/2022JD036728 for zonal means with less positive anomalies after around 2000, though it is rather a subtle change (particularly a decreasing trend in UTH 30-60ºS with positive anomalies before 2000).

The following is added to the revision in the paragraph of the Fig. 3 discussion:

Allan et al. (2022) examined changes in the anomaly characteristics in the zonal mean of AMIP 300–500 hPa RH, ERA5 300–500 hPa RH, and HIRS UTH (their Fig. 8a-c). Both the AMIP and HIRS time series showed a detectable decreasing trend in UTH 30°S-60˚S, and all three datasets showed decreasing amplitudes of anomalies after 2000. More specifically, AMIP and HIRS showed smaller positive anomalies while ERA5 exhibits smaller negative anomalies. The FIDUCEO MW UTH in Figure 3c of our study also shows subtly stronger UTH amplitudes before 2000, albeit with only a few years of data available. These changes after 2000 seem to be coincident with the decrease in the strength of El Niño events after 2000 as depicted by the MEI.v2, though such changes are not displayed in SST-only Niño indices such as the ONI.

Fig. 4/5 and 6/7 - a scatter of precipitation anomalies (possibly as % changes) verses UTH anomalies with some quantification of the relationship may be instructive and quite novel. It is not clear what the goal of the El Niño and La Niña comparisons are since these teleconnections are well known. If the goal is to

evaluate the differences between datasets it is not so obvious from these plots. La Niña minus El Niño may be another way to present similar information in half the number of plots.

Keeping both El Niño and La Niña figures helps readers visualize why there are negative UTH anomalies during El Niño when the tropical domain average is taken and positive UTH anomalies during La Niña. Having both El Niño and La Niña figures also helps the inter-comparison of the four UTH datasets during these major events specifically. Scatter plots (with histograms) of precipitation anomalies verses UTH anomalies, plus plots of SST and 200 hPa velocity potential verses UTH anomalies are added to the revision as new Figures 7-9:

To further assess the consistency of UTH datasets with several environmental variables, histograms of UTH anomalies vs. anomalies of GPCP precipitation, ERSSTv5 SST, and CFSR 200-hPa velocity potential during the peak six months of the 2015-16 El Niño are presented in Figures 7-9. The correlations between the anomalies of UTH and those of the three variables are also calculated and the correlation values are labelled at the top of each panel. Among the three variables, precipitation has the highest correlations with UTH (Figure 7), while SST has the lowest (Figure 8). both precipitation and velocity potential are proxies for vertical motion, so they are more directly tied to wet/dry UTH than the SST surface forcing. The increases of SST during El Niño events usually occur in the eastern-central Pacific, while the increases of both UTH and precipitation are more confined over the central Pacific. The UTH and precipitation fields both have a more balanced dipole between the central and western equatorial Pacific during a major El Niño, while the decrease of SST in the western equatorial Pacific does not match the strength of positive anomalies in the central-eastern equatorial Pacific. These patterns lead to overall higher correlations between UTH and precipitation than those between UTH and SST. The correlation values also illustrate that an SST-only ENSO index may not be as good of an indicator for the strength of UTH compared to an index that includes other environmental variables such as the MEI.

[Figure]

**Figure 7: Histograms of UTH anomalies of the four datasets vs. anomalies of GPCP precipitation during the peak six months of the 2015-16 El Niño. The blue line represents the linear regression line. The correlations between UTH anomalies and GPCP precipitation anomalies are labelled at the top of the panels.**

[Figure]

**Figure 8: Similar to Figure 7 except for UTH anomalies vs. ERSSTv5 anomalies.**

[Figure]

**Figure 9: Similar to Figure 7 except for UTH anomalies vs. anomalies of CFSR 200-hPa velocity potential.**

Among the UTH datasets, the MW data have higher correlations with the three environmental variables. The HIRS UTH correlation values are about 0.1 lower, primarily due to the lack of very humid anomalies in the infrared dataset. The histograms show that for all UTH datasets, the highest densities of anomalies are consistently centered around zero. The density of HIRS positive anomalies decreases rapidly beyond 5%, in line with the lowest ratio of large HIRS UTH shown in Figure 4d.

Fig. 8 suggests that CMSAF trends are inconsistent with the other datasets with increases in UTH over the whole region. Can this be quantified for the whole tropics and is this understood as the change in satellite offsets as implied. Was there no attempt to intercalibrate the satellite records by CMSAF?

The CMSAF UTH is based on a microwave humidity sounder FCDR generated by EUMETSAT. However, remaining inter-satellite discontinuities are noticed. The discontinuities in CMSAF UTH between satellites were documented in Lang et al. (2020).

Conclusions - some statements on the strengths and weaknesses of the datasets and possible issues identified would be welcome.

In addition to the strengths and weakness of the datasets discussed in the conclusion section (in terms of consistency and differences), a short paragraph is added at the end of the manuscript:

The infrared and MW UTH datasets have their own strengths and weakness. The HIRS dataset has the longest, over 43 years of observations so far, for long-term studies, and its variability, temporal phases, and spatial patterns are generally consistent with MW observations. However, being a clear-sky dataset, it does not capture the most humid regions. The MW datasets have a shorter time series, but they retain almost all-sky data, removing only the precipitating pixels, thus have a better sampling for a full spectrum of UTH especially for very humid data.

L324 - state that this is the CMSAF dataset which exhibits moistening trends relative to the other datasets

It is stated in the revision:

Wider spread of UTH moistening is observed in the CMSAF datasets.

Will future work make comparisons with reanalyses and CMIP6 models?

This would depend on the direction of the next phase of the GEWEX Water Vapor Assessment, and on whether there are sufficient changes in the UTH dataset versions. Current versions of two of the satellite UTH datasets have been compared with reanalyses and CMIP6 models in a few studies.

**Author response to Anonymous Reviewer #2**

This study inter-compared the polar-orbiting satellite UTH datasets by including four participating datasets, two of which are new datasets and two of which have updated versions. Further authors have presented case studies for the El Nino period in terms of time series and spatial anomalies analysis, and the authors also try to conclude that during both El Niño and La Niña events, the values of the spatial anomalies in the infrared dataset appear smaller than those in microwave datasets and the spatial patterns of the four datasets are generally consistent over the deep tropics. This information is very informative but readers of the remote sensing community or users of these datasets might be interested to know about the quality of these datasets for their further utilization. It is suggested that include the quality of the datasets as per your results and also mention the finding of any systematic biases over different geographical regions in the manuscript. In that respect the present study has potential for publication after incorporation of the comments/suggestions as given below:

Thank you for the comments and the helpful suggestions. In this study the quality of the data is assessed in terms of consistency with other UTH datasets. To improve on presenting the information, we added descriptions of changes over different geographical regions, and added more quantified discussions in the text and in the conclusions of the revised manuscript.

Comments:

The abstract is not informative enough.

More informative statements with quantified values are added to the Abstract in the revision:

During a major El Niño event, UTH had higher correlations with the coincident precipitation (0.60 – 0.75) and with 200 hPa velocity potential (-0.42 - -0.64) than with SST (0.37 - 0.49). Due to differences in retrieval definitions and gridding procedures, there can be a difference of 3-5% UTH between datasets on average, and more significant anomaly values are usually observed in the microwave UTH data. Nevertheless, the tropical-averaged anomalies of the datasets are close to each other with their differences being mostly less than 0.5% over tropical domain average, and more importantly the phases of the time series are generally consistent for variability studies.

The conclusion is not informative enough.

More informative statements with quantified values are added to the Conclusions in the revision:

The infrared UTH dataset exhibits the largest proportions of dry areas at the peak of El Niño and La Niña events (more than 4% larger ratio of dry areas compared to those of MW datasets). The MW datasets have larger proportion of humid measurements during El Niño events, while during a major La Niña such as the 2010-11 event, the ratios of humid areas are close to each other among three UTH dataset (differences less than 1%), except the CMSAF dataset which overall has larger humid areas.

During a major El Niño, the spatial correlations between UTH and SST are not high, with the correlation values in the range of 0.37-0.49. In the meantime, the spatial correlations between UTH and precipitation are higher, ranging in 0.60-0.75. The infrared dataset has lower correlation values (usually more than 0.1 smaller) with SST, precipitation, and 200 hPa velocity potential compared to those for the MW UTH datasets due to the smaller magnitude of anomalies in the infrared dataset.

Though there are apparent and expected differences in the values of total UTH due to differences in the definition and in the gridding procedure, the tropical-averaged anomalies of the datasets are close to each other (mostly within ±0.5% over tropical domain average), and more importantly the phases of the time series are generally consistent for variability studies.

No discussion about insitu measurements as well as reanalysis data that provides UTH in the introduction section (e.g. Radiosonde, etc.).

The following are added to the introduction section (in different paragraphs):

Measurement of UTH has traditionally been obtained from global radiosonde observations as part of the atmospheric water vapor profiles (e.g., Durre et al. (2018), Ferreira et al. (2019), Brogniez et al. (2015)).

The UTH datasets facilitated the evaluation of climate models and contributed to a better understanding of large-scale atmospheric processes (Allan et al., 2003; Soden et al., 2005; Chung et al., 2016; Allan et al., 2022; John et al., 2021).

The four datasets appear to be separated with two groups of similar UTH values. The values of CMSAF and FIDUCEO UTH are larger than the values of NCEI and UMIAMI UTH.

Why have authors taken the UMIAMI dataset as a reference for intercomparison? kindly add a magnitude of bias and also add some earlier analysis or findings with references.

The UMIAMI was used as a reference because it has the longest time period of AMSU-B and MHS. The conclusion on the relative differences of the datasets would be the same if a different dataset were used as the reference. The following shows a sample plot using the FIDUCEO UTH as a reference. Note that the SSM/T-2 UTH at the beginning of the FIDUCEO time series is not as stable as the AMSU-B/MHS data due to frequent missing data.

[Figure]

A reference that analyzed the relative differences among channel brightness temperatures of several sounders sensing the upper tropospheric humidity is added. The following is the revised text to clarify the choice of using UMIAMI as a reference:

To quantify the differences between datasets, the relative differences are calculated. Note that any of the four datasets can be used as a reference for this purpose. Among the MW UTH datasets, the UMIAMI dataset has the lengthiest time period of AMSU-B and MHS to allow for the longest MW comparison with others, and it is used as the relative reference in the calculation. Figure 1d shows that the anomaly values are mostly within ±0.5% UTH of each other, with the exceptions during El Niño events when the anomaly differences can be larger. Chung et al. (2016) analyzed the relative differences among the brightness temperatures of the channels sensing upper tropospheric humidity from HIRS, AMSU-B/MHS, and AIRS. The brightness temperature differences between the HIRS and AMSU-B/MHS were mostly within ±0.2 K.

Why did CMSAF show a negative anomaly during the year 2000 as compared to the other three datasets? Kindly explain.

This should be a result of remaining inter-satellite discontinuity in the channel brightness temperature data used for deriving UTH values. The resulting higher change rates during the common period was discussed later in the article as "The largest change rates are found in the CMSAF image, with positive changes covering most of the areas, consistent with the trend in Figure 1d. An earlier study (Lang et al.,

2020) plotted the time series of individual satellite's UTH from NOAA-15 to Metop-B for both FIDUCEO and CMSAF datasets (Figure 6 in that article). Their figure 6b showed that offsets between the UTH time series from consecutive satellite missions in the CMSAF record tend to be positive over time. When all the satellites are merged into one time series this may lead to a positive trend."

Line160: adds some suitable references.

This sentence is an observation of the time series patterns and is meant to say that the infrared clear-sky dataset has better sampling for dry regions than humid regions. It is revised to:

This indicates that the infrared clear-sky dataset may not fully capture the increase of water vapor during El Niño events due to the exclusion of very humid pixels associated with clouds, and tends to have better sampling of the dry regions.

Why does the NCEI UTH trend underestimate higher magnitude over the Tibetan Plateau as compared to the other three datasets?

The following discussion is added to the revision:

The clear-sky measurement excludes some high humidity data due to removal of cloudy pixels compared to MW datasets. The Jacobian of less-humid data has a lower peak in the atmosphere, and the lower tail of the Jacobian profile is closer to the surface (e.g., see Figure 1 in Brogniez et al. (2006)). Over a high elevation, increasing surface effect can be included in the observation radiances. A warming at the surface may contribute more to an infrared dataset due to larger portion of less-humid data.

Line304 Over the Indian Ocean, decreased UTH centered over the equatorial central Indian Ocean is surrounded with increased UTH in all datasets.

Such types of features are not seen in the CMSAF data plot and inconsistent trends are seen in the CMSAF data plot (Fig8). kindly explain.

We made a correction of the UTH changing rate description regarding the CMSAF data as below. Some explanation was given following the changing rate description as "The largest change rates are found in the CMSAF image, with positive changes covering most of the areas, consistent with the trend in Figure 1d. An earlier study (Lang et al., 2020) plotted the time series of individual satellite's UTH from NOAA-15 to Metop-B for both FIDUCEO and CMSAF datasets (Figure 6 in that article). Their figure 6b showed that offsets between the UTH time series from consecutive satellite missions in the CMSAF record tend to be positive over time. When all the satellites are merged into one time series this may lead to a positive trend."

Over the Indian Ocean, decreased UTH centered over the equatorial central Indian Ocean is surrounded with increased UTH in most datasets, except that the center of decreasing rates is confined to a smaller area around 15°S for the CMSAF UTH.

Reviewer's comments on "Assessing the consistency of satellite derived upper tropospheric humidity measurements" by Shi et al.

General comments:

This article describes an assessment of four UTH datasets: one using IR measurements and the other three using microwave (MW) measurements. The main purpose is to evaluate the consistency among these UTH datasets. Focus is placed on the tropics. Comparisons show that the four datasets are consistent in tropical-mean, interannual variability. For spatial patterns, they show broad consistency in ENSO-related action centers (e.g., Nino 4 region). Spatially, the IR-based UTH shows smaller variability than the MW-based UTH. The authors also re-examined a previous finding that was based on IR data (Shi et al. 2018), namely, upper tropospheric moistening during the El Nino events where convection is concentrated, but overall upper tropospheric drying when averaged over the whole tropics. They found that this conclusion is supported by the other three MW datasets. Finally, they analyzed the long-term changes as depicted in the four datasets by focusing on consistency among them.

Overall, I believe results from this assessment study should be useful to researchers in climate community who wish to use these datasets for diagnostic studies or model evaluations. It will contribute to the literature. As such, the paper is publishable. This kind of technical analysis fits the scope of AMT. Therefore, I'd recommend the paper be published after minor revisions.

We thank Reviewer #3 for the helpful comments and suggestions. We revised the manuscript as suggested. The following details the revision.

Specific comments:

L223-224: While UTH anomalies in the IR measurements are weaker than those in the MW in the Hovmoller diagram, the tropical-averaged UTH anomalies in the IR seem to be larger than those in the MW, as seen in Fig. 1(b). I see deeper dips in the IR data during the 1997-1998, and 2015-2016 ENSO events.

The Hovmöller diagram is averaged over 5°S-5°N, while the plots in Fig. 1b are averaged over 20°S-20°N. The deeper dips in the IR data during the 1997-1998, and 2015-2016 ENSO events mainly come from larger ratio of dry areas in the subtropics as shown in the new Fig. 4b in the revision. The following is edited and added to the paragraph discussing UTH spatial features during El Niño:

In the NCEI HIRS UTH panel, the magnitudes of both positive anomalies along the central-eastern equatorial Pacific and the negative anomalies in the western Pacific appear smaller than those in the other three microwave UTH panels, consistent with what is seen in the Hovmöller analysis discussed earlier.

However, over the tropical domain, the HIRS data have larger proportions of dry areas in the subtropics during El Niño events (resulting in larger overall dry area ratios shown in Figure 4b), leading to deeper dips of UTH during El Niño events displayed in Figure 1b.

L224 – 225: "Differences in the definition and computation of UTH, the sensitivity of different sounder, and clear-sky process may all contribute to the different strengths of derived anomalies". More details should be given to each of these causes as to how they affect the magnitude of the anomalies. This is a technical paper. The readers will likely care about such details.

This statement was referring to the observation of the weaker HIRS anomalies described in the previous sentence. We believe that the smaller magnitude of the anomalies compared to microwave data is primarily due to the definition used to computer UTH. The sentences are edited to:

In general, the equatorial UTH anomalies in the infrared measurements are relatively weaker than those in the microwave measurements. The definition used to compute the HIRS UTH may be the primary factor for the smaller magnitudes. The averaging of pixel-level brightness temperatures to the grids first before the UTH is computed may further smooth out the largest anomalies (both positive and negative).

L300: Since Nino 4 region has been mentioned and used for comparison in quite a few places, it makes sense to mark it up on the maps using rectangles.

The Nino 4 region is added using a rectangle to all the El Niño and La Niña maps.

L315 Conclusion: for a technical paper, I'd recommend the main findings be organized in bullet points to facilitate reading.

The main findings in the conclusion are organized in bullet form in the revision.

---

## Author Response (AR2)

We thank reviewer Richard Allan for reviewing our revised manuscript and providing helpful comments. We revised the manuscript accordingly. For clarity, reviewer's original comments are included in black, our responses are written in blue, and the revision in the updated manuscript is marked with green.

**Author response to Reviewer # 1, Richard Allan, on the revised manuscript**

Thank you to the authors for their careful resposnes. The new quantification is useful (I suggest using "(-0.42 to -0.64)" for ranges may avoid the problem of similarity with the minus sign) and the discussion of relationship between humidity, precipitation and other variables during ENSO is interesting. My only other comments are:

Thank you for the helpful comments and suggestions. All the ranges regarding correlation values are now written in the format of (value1 to value2).

1) the other part of how close is "close to each other" in the context of their use I meant was what accuracy is needed for the user (is 0.5% within uncertainty for example).

The GCOS required measurement uncertainty for the upper tropospheric humidity is 5% (https://gcos.wmo.int/en/essential-climate-variables/upper-vapour/ecv-requirements). While the differences among the several datasets of UTH anomalies are generally in the range of approximately 0.5%, indicating the likelihood of meeting the required uncertainty, the consistency analysis in this article does not address the absolute accuracy of the measurements. To do so, it would require comparison with measurements that can be traced to the International System of Units (SI), and it would be beyond the scope of our present study.

2) new material around Fig.4 - are the anomalies relative to a fixed multi-annual mean value over 20°S-20°N or for each grid point. Is it deseasonalised? Some more clarification would help. Since the paper is quite long now and this seems more complicated to explain and interpret than a simple proportion of low humidity and high humidity regions I recommend the new text and figure 4 and associated discussion is removed.

We think that the recommendation to produce the new figure (Figure 4) from the first round of reviews was helpful. In particular the figure adds more insight into why the tropical-domain averaged anomalies are negative during El Nino events by quantifying the proportion of negative and positive anomalies, and therefore, we'd keep the figure in the article. The anomalies are relative to each of the grid points and deseasonalized. We edited the text for more clarification as below:

To quantify the changing proportion of dry and humid regions derived from the different datasets, we calculate the percentage of grids with anomaly values greater or less than several selected values over 20°S-20°N (Figure 4). The anomalies are relative to each of the grid points and deseasonalized before the percentages are calculated.